SciPost Physics

Submission

# Free Growth Under Tension

Chenyun Yao[1] and Jens Elgeti[1★]

**1** Theoretical Physics of Living Matter, Institute for Advanced Simulation, Forschungszentrum Jülich, Jülich, Germany

★ j.elgeti@fz-juelich.de

## Abstract

Ever since the ground breaking work of Trepat et al. in 2009, we know that cell colonies growing on a substrate can be under tensile mechanical stress. The origin of tension has so far been attributed to cellular motility forces being oriented outward of the colony. Works in the field mainly revolve around how this orientation of the forces can be explained, ranging from velocity alignment to self-sorting due to self-propulsion. In this work, we demonstrate that tension in growing colonies can also be explained without cellular motility forces! Using a combination of well established tissue growth simulation technique and analytical modelling, we show how tension can arise as a consequence of simple mechanics of growing tissues. Combining these models with a minimalistic motility model shows how colonies can expand while under even larger tension. Furthermore, our results and analytical models provide novel analysis procedures to identify the underlying mechanics.

# 1   Introduction

Just as biochemical conditions, mechanics can affect the growth of biological tissues [1–5]. As the conjugate force to cell volume, particular attention has been given to pressure or tension. When a tissue grows, it exerts forces on its surroundings and vice versa experiences the reaction force. It is generally assumed [6] and experimentally confirmed [1], that pressure reduces growth. The idea is, that cells generate a pressure in order to expand in volume. In turn, the pressure exerted onto the tissue from the environment slows down this volume expansion. At the *homeostatic pressure*, growth is slowed down to the point where it equals the apoptosis rate: A steady state with constant cell turnover emerges. However Trepat et al. [7] showed that expanding cellular monolayers were not under pressure, but tensile stress. This tensile stress was attributed to cellular motility. Indeed, the two particle growth (2PG) model [8] extended by a velocity dependent activation and deactivation of a motility force was remarkably well able to explain the tensile growth [9]. The velocity dependent activation and deactivation of the motility force leads to an effective alignment interaction between the cell polarity and velocity, orienting the polarity outward, and thus generating tension. The model furthermore reproduced swirls in the bulk as found experimentally in confluent monolayers of Madin-Darby Canine Kidney (MDCK) cells [10, 11], and fingers at the advancing front reported for wound healing assays [12–15]. In 2021, Sarkar et al. [16] showed, that alignment interaction are not necessary to explain these. If the motility force just randomly reorients (Active Brownian Particle – ABP [17–19]), and the cell-cell adhesion allows for enough wiggle room, cells naturally sort with their polarity pointing outward due to the confinement [20, 21] and thus generating tension. That work however, did not include growth. In this work, we thus combine the 2PG model with the adhesion and motility model of Sarkar et al [16]. In short, cells consist of two point particles that repel each other by a constant force[1] of magnitude $G$. Upon reaching a critical distance, the cell divides. As in Sarkar et al [16], the cells interact with an extended Lennard-Jones potential of depth $\varepsilon$ and plateau width $\bar{\sigma}$, experience substrate friction of $-\gamma \boldsymbol{v}$, and possess a polarity $\boldsymbol{p}$ in which they exert a spontaneous motility force $F^M = \gamma v_0$. As in the Active Brownian Particle model, cells reorient by rotational diffusion. Quantities are reported in reduced units (indicated by $*$) with the units of length, time and energy corresponding to the cell size, cell turnover time, and the interaction strength of the reference colony. See methods for full details. Figure 1(a)(b) shows the surprising result: Even without motility force, the expanding monolayer develops a clear tensile stress! Adding motility (Fig.1(c)(d)) allows the colony to sustain higher tension, and also leads to fingers at the surface. Figure 1(e) highlights the key result whose details will be shown later: With the right parameters, our minimal model can perfectly match the experimental results of Trepat et al. 2009 [7]. Indeed, the phenomenon is rather robust, such that a rather broad range of parameters leads to very good agreement. In this work, we show how tension arises from the growth response to pressure, and how it is influenced by unregulated motility. Furthermore, we make strong predictions about the average cellular velocities that can be tested experimentally.

Simulations provide insights in the underlying mechanics. Figure 1 shows the superimposed snapshots of two different colonies from the simulations, as well as their local stress profiles. The non-motile colony grows as a circular disk, while the motile colony grows with irregular shape with finger-like protrusions. Both of these colonies are under tension, which builds up from the boundary and is the strongest in the center. Generally, we can distinguish four different phases of growth (see Fig.2): Slow and weakly growing cells form finite steady state colonies (I). Above a critical growth strength $G$ or motility $v_0$, the colony grows without reaching a steady state, either under tension (II) or under pressure (III)). In these cases,

---

[1]Earlier implementations of the 2PG model use a distance dependent growth force. In the spirit of minimal modelling, we now use a constant force.

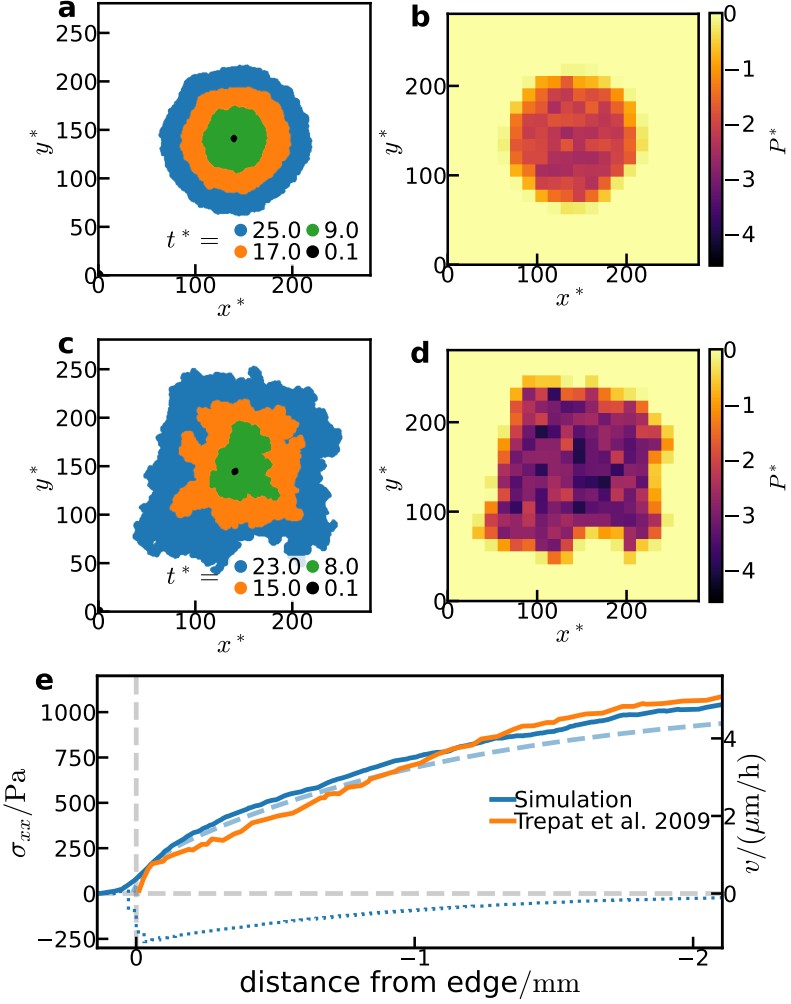

Figure 1: Growth of tensile colonies. (a) Superimposed snapshots of a growing colony of non-motile cells ($\varepsilon^* = 2, G^* = 16.4, v_0^* = 0$) at various times. It grows indefinitely in a roughly circular shape, but remains under tension. (b) The local stress profile corresponding to the last snapshot (blue, $t^* = 25$) of (a). (c) Superimposed snapshots of a growing colony of motile cells ($\varepsilon^* = 2, G^* = 15.0, v_0^* = 42.1$) at various times. It grows indefinitely, displaying fingering at the edge and large tension inside. (d) The local stress profile corresponding to the last snapshot (blue, $t^* = 23$) of (c). Both colonies are under tension, which is built up from the boundary to the center. (e) The stress profile of an expanding quasi-1d motile colony (blue solid) with $\varepsilon^* = 2, G^* = 17.8, v_0^* = 238.5$ matches those from Trepat et al. 2009 [7] by taking particle size $\sigma = 20\mu m$, cell turnover time $k_a^{-1} = 120h$, and interaction strength $\varepsilon = 2 \times 10^{-12}J$. With such choice of parameters, the motility is 40µm/h, the max retrograde flow is 1µm/h, the expansion speed is 0.1µm/h. The dashed and dotted lines are the corresponding theory prediction and velocity profile.

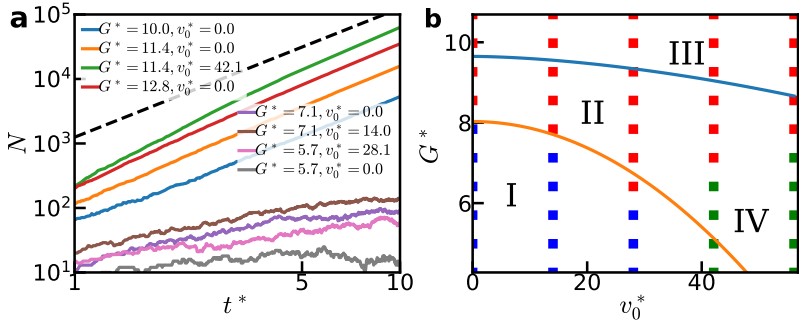

Figure 2: Phases of growth. (a) The number of cells $N$ versus time for some colonies. Some weakly growing colonies (lower four) only grow to a small finite stable size. For the colonies that grow indefinitely (upper four), $N$ grows quadratically in time in the asymptotic limit, whether it is motile or not. The dashed line indicates $N \sim t^2$. (b) The phase diagram of colony growth as a function of $G$ and $v_0$. Red points grow indefinitely, blue points grow to finite sizes. Green points indicate the scattered phase where the motility is too strong and cause particles to detach and scatter. See Supplementary Movies for examples of colonies in phase I through IV. The blue line is the contour line of $P_H = 0$ interpolated from simulation measurements. The orange line fitted from the simulations separate finite and infinite colonies. (See Supplementary Section S4 for the details of obtaining these two lines.) Between the two lines are the colonies that have negative $P_H$ but grow indefinitely. ($\varepsilon^* = 1$ in all simulations)

the number of cells grows quadratic in time, corresponding to a radial expansion at constant speed, as often observed experimentally [7, 22–24]. In all these phases, with low or moderate motility, cells do not detach from the colony. Only above a critical motility, the colony cannot be held together by the interactions anymore and cells behaves like a fast proliferating gas (IV). (See Supplementary Movies for examples of colonies in phase I through IV.)

## 2 Non-motile Quasi-1d Colonies

To understand how tension arises, one needs to keep in mind that cells have a tendency to proliferate more close to the boundary due to simple mechanical effects [1, 2, 25]. In essence, in order to grow, cells need to deform their surrounding. Close to the surface, the corresponding strain field is partially cut away, reducing the energetic cost of growth. On this basis, a quantitative understanding of phases I-III can be obtained from a simple analytical model. As in Ref. [2], we expand the local growth rate $k$ around the homeostatic pressure $P_H$, taking account of the additional pressure independent growth $\Delta k$ at the surface over a very thin width $\Delta x$ at boundary $x_0$:

$$k = \kappa(P_H - P) + \Delta k \Delta x \delta(x - x_0) \tag{1}$$

where $\kappa$ is the response coefficient, and the surface growth has been approximated by a delta function. The homeostatic pressure can be negative [25], leading to a negative bulk growth rate at zero pressure. The surface growth then leads to a stable steady state spheroid in three dimensions with a steady flux of cells from the proliferative rim to the apoptotic core [2, 25]. Friction with the substrate leads to an additional force on the cells, and thus can yield indefinitely growing tensile colonies. This mechanism can be best understood in a quasi-one-dimensional setup: The simulation box is chosen to be very large in $x$ direction and periodic in a short $y$ direction. The $y$ direction is short enough (about 10 cells) so it can be easily filled, but also large enough that cells can pass each other and form a continuous mass. Here we

restrict the analysis to the positive half space. Negative half space is given by mirror symmetry around the colony center at $x = 0$. The continuity equation than reads

$$\partial_t \rho + \partial_x(\rho v) = k\rho. \tag{2}$$

Assuming constant cell density $\rho$, Eq.(2) becomes

$$\partial_x v = k. \tag{3}$$

The simulations indicate, that the colony is homeostatically balanced in the $y$ direction ($\sigma_{yy} = -P_H$). With $P = -(\sigma_{xx} + \sigma_{yy})/2$ and defining $P_x = -\sigma_{xx}$ we get

$$k = \frac{\kappa}{2}(P_H - P_x) + \Delta k \Delta x \delta(x - x_0). \tag{4}$$

Force balance in $x$ reads

$$\partial_x \sigma_{xx} + f_{ext} = 0, \tag{5}$$

where $f_{ext}$ is the external force density. Without motility, the only external force on the monolayer is the background friction $f_{ext} = -2\rho\gamma v$. Combining Eq.(3), (4), and (5) in the bulk yields

$$\partial_x^2 P_x = \frac{1}{\lambda^2}(P_x - P_H), 0 < x < x_0 \tag{6}$$

where $\lambda^2 = (\rho\kappa\gamma)^{-1}$ (compare Ref. [26]). Because pressure is continuous, the boundary condition for the pressure reads $P_x(x_0) = 0$ (Note that the $\delta$ shaped surface growth does not cause a discontinuity in the pressure due to the two integrations involved, see Supplementary Section S1 for details.). Solving Eq.(6) yields

$$P_x = P_H(1 - \frac{\cosh(x/\lambda)}{\cosh(x_0/\lambda)}). \tag{7}$$

For $x_0/\lambda \gg 1$ and $x > 0$ we get

$$P_x = P_H(1 - e^{\frac{x-x_0}{\lambda}}), \tag{8}$$

i.e. the pressure builds up from the boundary into the bulk over a length scale $\lambda$, and reaches the homeostatic pressure $P_H$ deep in the bulk. The velocity profile is obtained from Eq.(5) and Eq.(8):

$$v = P_H\sqrt{\frac{\kappa}{4\rho\gamma}}e^{\frac{x-x_0}{\lambda}}, 0 < x < x_0. \tag{9}$$

The velocity thus also decays to zero exponentially. Note that for colonies with negative $P_H$ (contractile tissues), the velocity is negative for $0 < x < x_0$.

To calculate whether the tissue expands or shrinks, we integrate the growth rate (Eq.(4)) over the positive half space:

$$\frac{1}{2}\frac{dN}{dt} = \int_0^{x_0}\int_0^{L_y} \rho k\, dx\, dy = L_y(P_H\sqrt{\frac{\rho\kappa}{4\gamma}} + \rho\Delta k\Delta x). \tag{10}$$

The first term is the bulk contribution, which happens over a length scale of $\lambda$ from the boundary, and the second term is the surface growth contribution. Thus, the colony expands with a constant speed:

$$v(x_0) = \frac{1}{2L_y\rho}\frac{dN}{dt} = P_H\sqrt{\frac{\kappa}{4\rho\gamma}} + \Delta k\Delta x, \tag{11}$$

which is independent of the colony size. Thus, if the homeostatic pressure of the colony is positive, it will always grow indefinitely at a constant speed under pressure (Phase III). Below

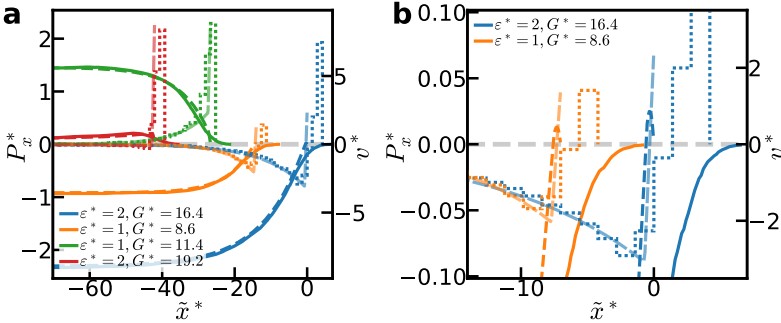

Figure 3: Pressure and velocity profiles of growing non-motile Quasi-1d colonies. $\tilde{x} = x - x_0 - x_s$, where $x_0$ is the surface position and the curves of different parameters are shifted with different $x_s$ for clarity. (a) The pressure profiles (solid) and velocity profiles (dotted) obtained from simulations of different parameters and the corresponding theory predictions (dashed). The pressure builds up from the boundary to $P_H$ in the bulk exponentially (except red, whose $P_H$ is positive but smaller than the pressure increase at the boundary due to surface growth). The velocity profiles decay exponentially in the bulk, and increase drastically but continuously at the boundary. For colonies with negative $P_H$, only the boundary is moving outward while the rest of the colony is moving towards the center. The theory predictions are calculated from the piecewise solution with all parameters measured independently (i.e. not fitted) and $\Delta x^*$ estimated to be 0.7. The theory and the simulations show good agreement. (b) The pressure profiles of two colonies with negative $P_H$ shown in (a) zoomed in at the front. The theory predicts an increase of pressure at the front for all expanding colonies, even if the homeostatic pressure is negative. However, this pressure is often too small to be observed from the simulations.

108 a critical (negative) homeostatic pressure of $P_H^C = -\Delta k \Delta x \sqrt{\frac{4\rho\gamma}{\kappa}}$, the surface growth can-
109 not compensate the total death in the bulk, and the colony will only grow to a finite size of
110 $2\lambda \tanh^{-1}(P_H^C/P_H)$ (Phase I), or shrink at a constant speed if the colony is initially larger. In
111 between, i.e. when $P_H^C < P_H < 0$, the colony will grow indefinitely under tension (Phase II),
112 where the tensile force is balanced by friction forces of the retrograde flow of cells from the
113 proliferating rim to the center.

114 Here, we treated surface growth as localized perfectly to the surface in a delta distribu-
115 tion, which leads to a discontinuous jump in the velocity at the surface (compare Eq.(9) and
116 Eq.(11)). A more rigorous piecewise solution (see Supplementary Section S1) displays a con-
117 tinuous velocity in the colony where the discontinuous jump becomes a linear increase over
118 the surface region, but otherwise converges to the solution presented above for $\Delta x/\lambda \ll 1$.

119 Eq.(11) predicts a constant expansion speed for all colonies in phase II or III and is linear in
120 $P_H - P_H^C$. Indeed, the simulations display a constant expansion speed linear in $G$ (see Fig.S7).
121 We use this constant expansion speed to obtain $\Delta k \Delta x$. Obtaining the homeostatic pressure
122 and other bulk tissue properties from bulk simulations (i.e. without a fit, see Supplementary
123 Section S5), and estimating $\Delta x^* = 0.7$ reproduce the simulation data of pressure and velocity
124 remarkably well (Fig.3).

125 For an expanding front, the pressure rises at the boundary, even though the value of this
126 pressure may be too small and narrow to be observed in the simulations. So for colonies
127 with $P_H < 0$, from the boundary to the bulk, the pressure first increases but then peaks, and
128 decreases to $P_H$. This also applies when the $P_H$ is positive but smaller than the pressure built
129 at the boundary (see for example the red curve in Fig.3). In these cases, the velocity profile
130 is negative, i.e. a retrograde flow of cells moving inward, except in a small region near the

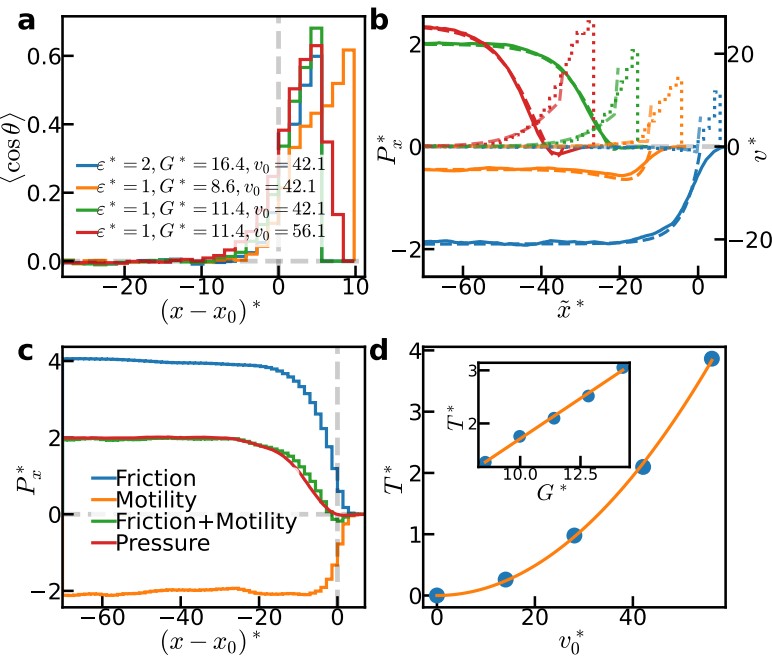

Figure 4: Polarization, pressure, and velocity profiles of motile quasi-1d colonies. $\tilde{x} = x - x_0 - x_s$, where $x_0$ is the surface position and the curves of different parameters are shifted with different $x_s$ for clarity. (a) The polarization profile of different colonies. The polarization has a sharp distribution at the boundary of the colony. The maximum polarization is roughly independent of the parameters. (b) The pressure and velocity profiles of different motile colonies (colors as in (a)). The solid and dotted lines are the pressure and velocity profiles measured from simulations respectively. The dashed lines are calculated from the theory without adjustable parameters. The theory predictions matches the simulation results well. Motility indeed generates tension at the boundary of the colonies, though the pressure still goes to $P_H$ in the bulk. (c) Motility force and friction build stress inside the colony. They add up to the pressure. (d) The tension generated by motility $T$ is quadratic in $v_0$ and linear in $G$ (inset).

boundary. It is this retrograde flow, that balances the tension inside the colony. Thus our simulations and analytical arguments predict that for non-motile tissues that display tension, cells should exhibit a retrograde flow.

## 3 Motile Quasi-1d colonies

Without motility, a tensile homeostatic stress is balanced by friction due to a flux of cells inwards from the proliferating boundary. Motility adds a second external force to the force balance equation, as cells exert their motility force $f_a = \gamma v_0$ in direction $\boldsymbol{p} = (\cos\theta, \sin\theta)$, where $\theta$ is the angle between the single cell polarization and the $x$ axis.[2] The force balance equation thus reads

$$\partial_x \sigma_{xx} - 2\rho\gamma v + 2\rho\gamma v_0 \langle\cos\theta\rangle = 0, \tag{12}$$

---

[2]Note that freely moving cells exert no net force on the substrate, as their motility force is exactly balanced by their friction force.

with the average polarization $\langle \cos \theta \rangle$. Fig.4(a) shows, that in our simulations the polarization is zero in the bulk, but increases sharply towards the boundary, similar to non-proliferating motile colonies [16]. To model the motile colonies, we assume phenomenologically the distribution of the motility force density to be an exponential function of $x$: $2\rho \gamma v_0 \langle \cos \theta \rangle = F e^{\frac{x-x_0}{\lambda_m}}$. By following similar procedures as above, we obtain the pressure profile for a quasi-1d motile colony:

$$P_x = P_H (1 - e^{\frac{x-x_0}{\lambda}}) + T \frac{\lambda^2}{\lambda^2 - \lambda_m^2} (e^{\frac{x-x_0}{\lambda_m}} - e^{\frac{x-x_0}{\lambda}}), \tag{13}$$

where $T = \int F e^{\frac{x-x_0}{\lambda_m}} \, \mathrm{d}x = F \lambda_m > 0$ is the total tension generated by the motility force over a length scale $\lambda_m$. Motility generates tension at the boundary, but the pressure still builds towards $P_H$ in the bulk over a length scale of $\lambda$ (in the simulations $P_H$ and $\lambda$ depend on $v_0$). Fig.4(b) compares the measured pressure profiles to the analytical expression without adjustable parameters (parameters determined from the orientation profile and independent simulations, see Supplementary Section S5), and also displays the measured velocity profiles for different colonies. Note that for motile colonies the interface roughens, leading to less accurate agreement.

Comparing simulations with different motility reveals two effects: On the one hand, motility also favors bulk growth, thus increasing the homeostatic pressure. Indeed, we find that the homeostatic pressure increases nearly quadratically with motility force (Fig.S2).

On the other hand, motility generates tension $T$ at the leading edge as predicted by Eq.(12). This tension can dominate the pressure profile. Even for positive homeostatic pressure, we observe a dip into tension close to the edge, and for small negative homeostatic pressure, the tension overshoots before relaxing back to the homeostatic pressure. The larger the motility force, the stronger this effect. Importantly, this motility also effects the velocity profile, masking, or even inverting the retrograde flow observed for non-motile tensile colonies.

The simulations allow us to directly access the different contributions to the pressure in the colony. By measuring friction and motility forces from the simulations and integrating them from the stress-free surface, we obtain the tension $T$ generated by motility and the pressure build up due to friction forces. Consistently, they add up to the total pressure measured via the virial. Fig.4(c) shows these contributions for one exemplary case. Fig.4(d) and its inset shows the relationship between the total tension generated by motility $T$ as a function of motility $v_0$ and growth force $G$. The total tension is observed to be quadratic in $v_0$ and linear in $G$.

To see how the motility induced tension supports tensile colonies, we obtain the expansion speed by integrating the growth rate as above:

$$\begin{aligned} v(x_0) &= \frac{1}{2\rho\gamma} \left( \frac{P_H}{\lambda} + \frac{T}{\lambda + \lambda_m} \right) + \Delta k \Delta x \\ &= \left( P_H + T \frac{1}{1 + \frac{\lambda_m}{\lambda}} \right) \sqrt{\frac{\kappa}{4\rho\gamma}} + \Delta k \Delta x. \end{aligned} \tag{14}$$

The expansion speed is still a constant, but with an additional contribution due to motility induced tension. Fig.5(a) shows the simulation data of the expansion speed vs motility $v_0$ (also see Fig.S10). The expansion speed is quadratic in $v_0$ as expected from the $v_0 \rightarrow -v_0$ symmetry of our model. According to Eq.(14), the critical homeostatic pressure becomes

$$P_H^C = -\Delta k \Delta x \sqrt{\frac{4\rho\gamma}{\kappa}} - T \frac{1}{1 + \frac{\lambda_m}{\lambda}}, \tag{15}$$

thus expanding the phase of indefinitely growing tensile colonies to even larger tension (Phase II, see Fig.5(b)). One might expect that sufficiently high motility can thus sustain an arbitrarily

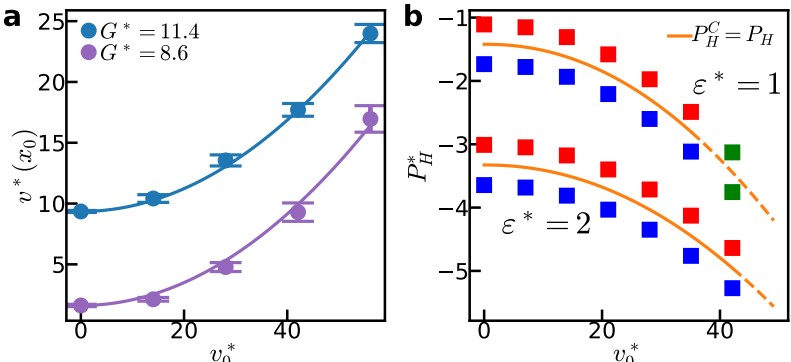

Figure 5: Effects of motility on the growth of motile quasi-1d colonies. (a) The colony expansion speed $v(x_0)$ of colonies with $G^* = 11.4$ and $G^* = 8.6$ as a function of the motility $v_0$. The fit with the form $v(x_0) = v(x_0)|_{v_0=0} + C v_0^2$ shows that the expansion speed is quadratic in $v_0$. Error bars indicate standard deviations. (b) The maximum homeostatic tension $(-P_H^*)$ at which the colony can still grow indefinitely (orange line). The squares correspond to simulated colonies that grow indefinitely (red) and to a finite size (blue). Beyond certain $v_0$, the colony becomes scattered (dashed line and green squares). Colonies with larger $\varepsilon$ are more resistant to detaching for the same expansion speed.

high homeostatic tension. However, as motility increases, adhesion is overcome, and cells can detach from the main colony and a proliferative gas is formed. In order to reach even more tension, higher adhesion forces are needed. Increasing the interaction strength $\varepsilon$ has two effects: For one, it increases the surface growth effect (first term in Eq.(15)). Second, it keeps the tissue cohesive at even higher motility forces (Fig.5(b)).

## 4  Two dimensions - growth on a substrate

If the shape of the colony does not deviate too much from a circle, the above analysis can be applied to two dimensional growth with radial symmetry (see Supplementary Section S2 and Fig.S1). Indeed, the solution for a circular geometry converges to the one dimensional case for radii much larger than $\lambda$ (Fig.6(a)). Importantly however, as shown in Fig.1, motility creates fingers at the boundary. We observe, that a finger is caused by the boundary accumulation of outward-polarized particles, and that fingering is stronger if the equivalent colony without motility grows slower. Furthermore, fingers increase the surface area and thus the growth due to surface growth. This is partially enhanced by the effect that daughter cells inherit the motility polarizations of their mother cell resulting in a positive feedback loop. Without this polarization heritage, fingering is reduced and the colony grows slower (see Supplementary Video 3). Given this fingering tendency, one might expect fractal growth of the colonies. However, as Fig.6(b) shows, while the number of boundary cells is increased, it still scales asymptotically with the square root of the total number of cells - indicating against fractal properties. Despite all this, Fig.6(a) shows that the expansion speeds of motile 2d and quasi-1d colonies are very close, evidencing that the effect of fingers on the overall growth are minor.

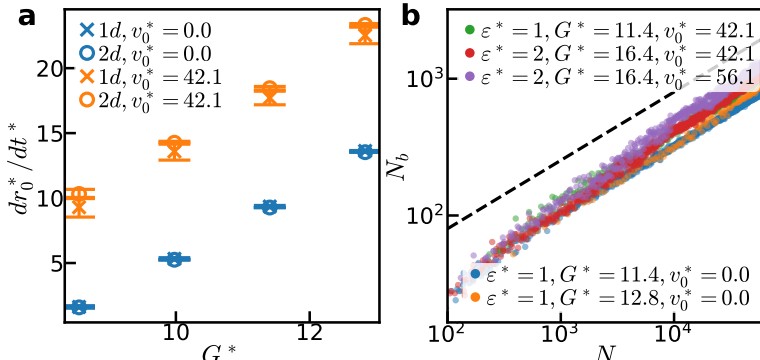

Figure 6: The growth of 2d colonies. (a) The comparison of the expansion speeds of quasi-1d ($dx_0/dt$) and 2d ($dr_0/dt$) colonies. They show good agreement, even with motility. The error bars indicate standard deviations. (b) The relationship between the number of boundary cells $N_b$ and the number of cells $N$ (see Supplementary Section S6 for the determination of boundary cells). The dashed line indicates $N^{\frac{1}{2}}$. For non-motile colonies, the boundary cell number behaves exactly as $N^{\frac{1}{2}}$. When a motile colony is small, fingers cause fractal-like behaviour. But at the limit of large colony, the exponent still goes to 1/2. So the fingers become an undulation of a fixed amplitude, increasing the roughness of the surface. The larger the motility, the larger the roughness.

## 5 Conclusions

In this paper, we explore the mechanics of growth of cell colonies on a substrate. Importantly, we find four different phases of growth: In Phase I, the colony is so contractile it can only grow to finite size. As the homeostatic pressure increases above the critical pressure determined by Eq.(15), the colony grows indefinitely, while remaining under tension (Phase II). The tissue always reaches its homeostatic state in the center, thus becoming under pressure, once the homeostatic pressure turns positive (Phase III). Finally, for very strong motility, groups of cells detach, and we arrive at a growing gas (Phase IV). The ability to grow indefinitely while under tensile stress is enabled by two factors: 1. the propensity of cells to grow faster at the interface, and 2. outward directed cellular motility. These two factors can act independently, or act in consort to balance an even greater tensile core. However, in our simulations the additional growth at the surface is always present, as it arises from mechanical principles. Using continuum theory, we quantitatively predict the transition between Phase I,II, and III. The critical pressure consequentially has two contributions: First, excess growth at the interface results in an retrograde flow of cells, which, due to friction, can balance out a tensile core. Second, motility in combination with self-generated outward polarization of cells results in tension buildup that can additionally balance a tensile core.

Our findings can help interpret experimental findings of tensile colony growth [7, 24, 27, 28]. Combining these traction force experiments with measuring cellular velocities (for example via particle imaging velocimetry [10, 11, 24, 29, 30]) may help to gain further insight into the underlying mechanics. Especially, the presence of a retrograde flow of cells would be very interesting, even though our model suggests it could be absent due to cellular motility.

Another key prediction that can be tested experimentally is that the pressure in the colony will always relax toward its homeostatic pressure over a length scale of $\lambda$, and the center of a large colony will always be under its homeostatic pressure, which can indeed be observed in experiments [28]. Apoptosis, or other forms of cell loss, are required for the homeostatic pres-

sure to be negative, which means it is also required for a growing colony to be under tension in the center (See Fig.S11 for the effect of apoptosis rate and adhesion on the homeostatic pressure). In reality, this corresponds to colonies that are naturally contractile. For example, in Ref. [31, 32], the tissues retract after being cut by laser, indicating they are under tension.

Tension in expanding colonies can also be generated by outward pointing motility and a viscous stress model [33, 34]. However, if the growth is not coupled to pressure, central stress in the colony is zero in the large colony limit (e.g. take $L \to \infty$ and $x \to 0$ in Eq.(5) of ref. [33]), while our model predicts a finite homeostatic tension even for indefinitely growing colonies.

Our model has some important shortcomings however. For one, it does not consider any form of motility alignment, even though biological cells certainly do not reorient randomly [27, 29, 35–37]. While useful to uncover fundamental principles about how tension may arise, a quantitative matching of experimental data will certainly require some form of alignment. Our model was able to quantitatively match the data of Trepat et al. [7], however also suggests a retrograde flow, which has not been reported. Adding motility alignment could remove this retrograde flow, and a detailed quantitative comparison of traction and velocity maps may help uncover the true underlying mechanism. Similar to Ref. [37], we want to implement various motility alignment mechanisms and compare them to experimental data in future works.

Second, part of the force balancing tension in the colonies center comes from friction. Thus the type of friction plays a key role. Here, we assumed simple linear friction ($f = -\gamma v$) while cells can certainly display a more complex behavior like dry friction or even an active response to external force.

Third, we have not explored the finger formation in detail. We expect that similar to competing tissues, linear stability analysis [38] could shed a light on how these fingers form. Subsequent comparison to simulations can than reveal further insights [39].

Finally, leader cells, supra-cellular actin cables and their interactions certainly play a role in real MDCK colonies [12–14] and possibly other cell lines [15, 40]. This work can only paint the generic picture of how mechanics of tensile growth can function – a detailed quantitative comparison will need to take details of the specific cell line into account.

**Funding information**   CY is partially sponsored by China Scholarship Council (202108310086).

**Supplementary material**   Supplementary information, supplementary movies, source code, processing scripts, and simulation data are available at zenodo.org (doi:10.5281/zenodo.15187628).

## A   Simulation Method

We use the 2PG model [8] for tissue growth. In the model, each cell consists of two particles with diameter $\sigma$. The two particles are repelling each other with a constant growth force $G$. When the distance between the two particles exceeds a threshold $r_c$, the cell divides, and a new particle is placed close to each old particle to form two new cells. Cell apoptosis is modeled as a constant cell removal rate $k_a$. We implement a Dissipative Particle Dynamics (DPD) type thermostat for intracell and intercell particle interactions. The interaction includes a dissipative force

$$\boldsymbol{F}_{ij}^{D} = -\gamma_D \omega^D(r_{ij})(\boldsymbol{v}_{ij} \cdot \hat{\boldsymbol{r}}_{ij})\hat{\boldsymbol{r}}_{ij} \tag{A.1}$$

and a random force

$$\boldsymbol{F}_{ij}^{R} = \mu \omega^R(r_{ij})\varepsilon_{ij}\hat{\boldsymbol{r}}_{ij}. \tag{A.2}$$

Here, $\boldsymbol{v}_{ij} = \boldsymbol{v}_j - \boldsymbol{v}_i$, $\varepsilon_{ij}$ is a Gaussian variable with zero mean and unit variance, $\omega^D(r_{ij})$ and $\omega^R(r_{ij})$ are weight functions, $\gamma_D$ is the friction coefficient, which can be chosen independently for intercell and intracell interaction, and $\mu$ is the strength of the random force. To fulfil the fluctuation-dissipation theorem, $\mu^2 = 2\gamma_D k_B T$ and $\omega^D(r_{ij}) = [\omega^R(r_{ij})]^2$ must be satisfied. For intracell particle interaction, we choose $\omega^D(r_{ij}) = 1$. And for intercell particle interaction, we choose $\omega^D(r_{ij}) = (1 - r_{ij}/R_{PP})^2$, where $R_{PP}$ is the cutoff radius.

For the motility and interaction, we incorporate the modified ABP model of Sarkar et al. 2021 [16]. In this model, particles not belonging to the same cell interact with each other with the extended Lennard-Jones (LJ) potential:

$$V_{\text{ELJ}}(r) = \begin{cases} 4\epsilon[(\dfrac{\sigma}{r})^{12} - (\dfrac{\sigma}{r})^6], & r < 2^{\frac{1}{6}}\sigma \\ -\epsilon, & 2^{\frac{1}{6}}\sigma \leq r < 2^{\frac{1}{6}}\sigma + \bar{\sigma} \\ 4\epsilon[(\dfrac{\sigma}{r-\bar{\sigma}})^{12} - (\dfrac{\sigma}{r-\bar{\sigma}})^6], & 2^{\frac{1}{6}}\sigma + \bar{\sigma} \leq r \end{cases} \tag{A.3}$$

where $\epsilon$ is the interaction strength, $\sigma$ is the diameter of the particle, and $\bar{\sigma}$ is the width of the extended basin, which we choose to be $0.3\sigma$. Additionally, each particle is subject to a propelling motility force with a constant magnitude $F^M = \gamma v_0$, where $\gamma$ is the background friction coefficient. The direction of the motility force is identical for both particles constituting the same cell, and undergoes a rotational diffusion

$$\dot{\theta}_i = \sqrt{2D_R}\eta_i^R,$$

where $D_R$ is the rotational diffusion constant and $\eta_i^R$ is again a Gaussian white noise. After a division, the two daughter cells inherit the motility polarization of the mother cell. Even though the origin of the rotational diffusion can be athermal since this is an active system, we still set the relationship between $D_R$ and $D_T$ to satisfy the Einstein relation $D_T = k_B T/\gamma = D_R \sigma^2/3$. This model contains neither motility alignment nor leader cell mechanisms.

In summary, the total equation of motion of a particle $i$, with $k$ the other particle of the same cell, is

$$m\ddot{\boldsymbol{r}}_i = \boldsymbol{F}_{ik}^{G} + \boldsymbol{F}_{ik}^{D} + \boldsymbol{F}_{ik}^{R} + \sum_{j \neq i,k}(\boldsymbol{F}_{ij}^{ELJ} + \boldsymbol{F}_{ij}^{D} + \boldsymbol{F}_{ij}^{R}) + \boldsymbol{F}_i^{BD} + \boldsymbol{F}_i^{BR} + \boldsymbol{F}_i^{M}, \tag{A.4}$$

where each term on the right hand side means growth force, intracell dissipation, intracell fluctuation, intercell extended LJ interaction, inter cell dissipation, intercell fluctuation, background friction, background fluctuation, and motility force respectively. Equations of motion are integrated with a velocity-Verlet algorithm with an additional calculation of dissipative

291 forces (DPP-VV from Ref. [41]).

292 Physical quantities are reported in reduced units, indicated by an asterisk. We use the diame-
293 ter of the particles $\sigma$, the cell turnover time $k_a^{-1}$, and the interaction strength of the reference
294 tissue $\varepsilon = 1$ as the reference parameters.

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
