# Peer review of "Free Growth Under Tension"

_SciPost Physics, doi:SciPost Phys. 19, 097 (2025)_

## Round 1 · Referee Report · Anonymous (Referee 1) · 2025-6-17

Strengths

  1. Incorporates cell growth and death in an active matter framework
  2. Well-established results using numerical simulations and theoretical analysis

Weaknesses

  1. Would be improved with a clearer comparison, highlighting differences, of the model with only motility.

Report

This paper models the effects of growth in cell colonies. They study the stress and flow profiles of growing cell colonies with and without motility, showing that cell monolayers can be under tension even in the absence of cell motility.

The results are interesting to an active matter audience. I would be happy to recommend this paper for publication, provided the authors can clarify the following points:
1. The origin and manipulation of the \delta function in Section I is not clear. What happens at x = -x0 – is there an additional assumption of symmetry made here? This edge growth term is ignored in Eqs. (5)—(9) when calculating the velocity and pressure profiles. Why does this term have no effect on the boundary conditions, e.g. P_x(x_0) = P_x(-x_0) =0.
2. What boundary conditions are used for the velocity field solution in Eq. (9)? It would be helpful to present the complete form of the full velocity, before taking the x0/\lambda>>1 approximation.
3. In Section II, it is not clear to me how the polarity p is defined. In particular, where does \theta come from? What are the dynamics of the polarity field?
4. It would be helpful to have figures to identify or characterize the different phases in Fig. 2. In particular, phase (IV) is not shown or discussed. How are the boundary lines in Fig 2b delineated?
5. What are the key differences predicted for the observable stress and velocity profiles for the proposed model with growth, versus other commonly used models with only motility e.g. [1]? I found a clear comparison missing. What experiments would you suggest to test this model?
6. What is the intuitive explanation for having tension in a growing monolayer?
7. In addition, the paper is littered with multiple typos. A few (sometimes recurring) examples: ‘kenotaxis’, ‘detph’, ‘repell’, ‘veloicty’, ‘optianed’, ‘boudnary’, ‘separate’, ‘negatiev’, ‘enabeled’, ‘shortcommings’, ‘usefull’, ‘quais’. Also a few grammar errors e.g. ‘it is’ instead of ‘its’ in Line 45.

[1] Effective viscosity and dynamics of spreading epithelia: a solvable model. Blanch-Mercader, C. , Vincent, R. , Bazellières, E. , Serra-Picamal, X. , Trepat, X. and Casademunt, J. Soft Matter, 2017,13, 1235-1243

Requested changes

See above

Recommendation

Publish (easily meets expectations and criteria for this Journal; among top 50%)

  • validity: high
  • significance: high
  • originality: high
  • clarity: good
  • formatting: good
  • grammar: acceptable

Author:  Jens Elgeti  on 2025-07-29  [id 5691]

(in reply to Report 1 on 2025-06-17)
Category:
answer to question

This paper models the effects of growth in cell colonies. They study the stress and flow profiles of growing cell colonies with and without motility, showing that cell monolayers can be under tension even in the absence of cell motility.

The results are interesting to an active matter audience. I would be happy to recommend this paper for publication, provided the authors can clarify the following points:

We kindly thank the referee for carefully reading our manuscript and providing constructive feedback. We are happy to clarify the points raised by the referee and corrections in the following:

  1. The origin and manipulation of the \delta function in Section I is not clear.

Indeed, this should have been made clear. The delta function of the growth rate comes from the additional growth rate that appears in a very thin region at the surface. This region is very thin compared to other length scales. Thus in the simple solution presented in the main text we take the surface growth as a delta function. We added this clarification after Eq.1.

What happens at x = -x0 – is there an additional assumption of symmetry made here?

Indeed, we should have pointed out that we assume mirror symmetry with respect to x=0. We mention this now explicitly before Eq.2.

This edge growth term is ignored in Eqs. (5)—(9) when calculating the velocity and pressure profiles. Why does this term have no effect on the boundary conditions, e.g. P_x(x_0) = P_x(-x_0) =0.

The referee is right to point out this apparent discrepancy that also confused us initially. Indeed, this is one reason for the piecewise calculation in the SI. This piecewise solution shows how the delta function in the growth rate disappears from the pressure equations (Eq.5-9) due to the two integrations involved. In essence, the delta function in growth rate corresponds to a finite step in the velocity profile. The effect on the stress however is proportional to the velocity (finite) times its width (->0). Thus the effect on the stress by a delta-shaped growth rate is zero. Please see Supplementary Section S1 for further details. To facilitate this point to the reader, we added a note to the main text, and a more detailed explanation in Supplementary Section S1.

  1. What boundary conditions are used for the velocity field solution in Eq. (9)?

The velocity field is directly obtained from the derivative of the pressure profile, so no additional boundary condition is needed. Tough zero velocity at center is equivalently implied by the symmetric boundary condition of pressure P(x_0)=P(-x_0)=0. Note that eq. 9 is limited to 0<x<x0, i.e. not including x0. Due to the delta-growth, the velocity is discontinuous at x=x0, as shown in Eq.11. This is now also highlighted in the manuscript.

It would be helpful to present the complete form of the full velocity, before taking the x0/\lambda>>1 approximation.

We agree it is useful to present a full solution without taking the limit of \Delta x to zero. However, the complete form is rather complicated and does not contain much further insight. We thus decided to present it in the SI, and only present the simple solution that already contains all the relevant physics in the main text. We added a note pointing to the solution in the SI at the corresponding point of the main text.

  1. In Section II, it is not clear to me how the polarity p is defined. In particular, where does \theta come from?

The single cell polarity is defined as a unit vector p=cos\theta \hat{x} +sin\theta \hat{y}, where \theta is the angle between the polarity of a single cell and the x axis. Since it's quasi-1d, by taking the local average, the y component cancels out, and we use <cos\theta> to refer to the local coarse grained 1d polarity. We now provide this definition of \theta explicitly .

What are the dynamics of the polarity field?

We only impose random reorientation of individual cells. We do not know explicitly the dynamics of the polarization field. We can only measure the stable distribution, and phenomenologically assume it's an exponential distribution. We emphasize now this phenomenological nature.

  1. It would be helpful to have figures to identify or characterize the different phases in Fig. 2. In particular, phase (IV) is not shown or discussed.

We agree it would be helpful. Movies showcasing the different phases have been added to the SI.

How are the boundary lines in Fig 2b delineated?

The boundary lines are explained in the captions of Fig. 2(b), but indeed not detailed enough. The line for P_H=0 (Blue) is obtained from a fit for P_H with a phenomenological form C_0+ C_1 G + C_2 v_0^{C_3} (The validity of this form is shown in Figure S2). Then we set P_H=0 to obtain a relation between G and v_0. The line for the expansion speed v(x_0)=0 (Orange) is obtained similarly (C_3 = 2 in this case). We added this explanation to the SI.

  1. What are the key differences predicted for the observable stress and velocity profiles for the proposed model with growth, versus other commonly used models with only motility e.g. [1]? I found a clear comparison missing. What experiments would you suggest to test this model?

The referee is right to point out that we lack a comparison with other models studying the same problem. The most characteristic result of the model for a non-motile colony under tension is the retrograde flow in the bulk. So experimentally one can check if such retrograde flow exists in tensile non-motile cell colonies. For motile colonies (especially with alignment) though, retrograde flow is not needed. Another characteristic is that the stress is a constant beyond a certain length from the interface. Over a length scale lambda, it relaxes towards the homeostatic pressure, which should be observable in large enough cell colonies. The above points can be used to test this model experimentally. In comparison with other models that only include motility and viscous stress like Ref. [33] and [34], we obtain similar velocity profiles, but the most important difference lies in the pressure profile: In the large colony size limit, their stress in the center of the colony would be zero, while our stress in the center is the non-zero homeostatic stress. The above argument has been added to the discussion.

  1. What is the intuitive explanation for having tension in a growing monolayer?

This question can be broken into 4 parts:

(i) What is homeostatic tension?

It means that in the homeostatic state where the total growth in the tissue is balanced by its total apoptosis, the tissue must be under tension. We call this tissue to have negative homeostatic pressure.

(ii) What is tension in a monolayer?

This corresponds to tissues that are contractile, which can be seen in experiments. For example, in Ref.[31,32] the tissues retract after being cut with laser, indicating they are under tension.

(iii) How is this tension balanced?

On single cell scale, each cell receives the cell-cell interaction forces from two sides and a traction force connected to the surface. The tension built up is a result of force balance on each row of cells (tug-of-war). On colony scale, the tensile bulk is balanced by the friction and/or motility forces with the surface.

(iv) How can it still grow under tension?

Due to background friction, the stress in the colony will relax toward homeostatic tension into the bulk. As a result, net growth is only negative in a finite region behind the front. On the other hand, cells have additional growth rate at the surface. If this surface growth is larger than the total loss over the finite region behind the front, then the colony will grow under tension.

We expanded the discussion reflecting these points.

  1. In addition, the paper is littered with multiple typos. A few (sometimes recurring) examples: ‘kenotaxis’, ‘detph’, ‘repell’, ‘veloicty’, ‘optianed’, ‘boudnary’, ‘separate’, ‘negatiev’, ‘enabeled’, ‘shortcommings’, ‘usefull’, ‘quais’. Also a few grammar errors e.g. ‘it is’ instead of ‘its’ in Line 45.

We are sorry for all the typos, and we kindly thank the referee for pointing them out. Typos have been corrected in the manuscript.

Reference [1] Effective viscosity and dynamics of spreading epithelia: a solvable model. Blanch-Mercader, C. , Vincent, R. , Bazellières, E. , Serra-Picamal, X. , Trepat, X. and Casademunt, J. Soft Matter, 2017,13, 1235-1243

Thank you for pointing out this relevant work, it is now cited as reference [33]

---

## Round 1 · Referee Report · Anonymous (Referee 2) · 2025-7-1

Report

The manuscript by Yao and Elgeti investigates the emergence of mechanical tension in expanding cell colonies growing on substrates, challenging the conventional assumption that tension arises solely from outward-directed cellular motility. Using a combination of particle-based tissue-growth simulations and analytical modeling, the authors demonstrate that tension can spontaneously develop even without motility, driven by mechanical boundary effects promoting growth near colony edges. Incorporating a minimal motility model further amplifies tension and reproduces experimental observations, such as finger-like protrusions at colony fronts. Their findings provide a novel framework for distinguishing different mechanical growth regimes and predict measurable retrograde cellular flows, offering new interpretations for experimental systems.

The manuscript is generally well-written and provides interesting perspectives. However, there are some issues regarding presentation and the interpretation of the results which should be addressed before publication.

Disclaimer: We could not find the supplement on SciPost and looked at the arXiv version of the supplement assuming it would be similar.

Major comments

  1. Definition of the model and initial presentation of effects: While the model might be very well-known and intuitive for the authors, the beginning of the manuscript is not easy to digest for outside readers: While a few model components are mentioned in the main text, the entire definition is outsourced to methods and the supplement. The main text should at least give the necessary intuition regarding all relevant model mechanisms (apoptosis, adhesion, how does homeostasis arise in this model?) before jumping to the first results (around line 39). In addition, the presentation of the initial results is a bit superficial (Fig. 1), it is not clear how the pressure is measured (although explained in the supplement), etc. It might help to first finish a slightly more expanded model introduction and then switch to a proper presentation of the results.

  2. Model transparency and choice of main parameters: The significance of certain modeling details is not always clear. If we understand correctly, the growth force in Refs. 8 (on which the model is based) and 9 is not constant but depends on the distance between cell particles. Also, a different kind of repulsion is used. What is the rationale behind these changes? Do they alter the results? Moreover, the manuscript uses the microscopic parameters growth force $G$ and adhesion strength $\epsilon$ to explore different growth regimes ($\epsilon^*=2$ in Fig. 1, $\epsilon^*=1$ in Fig. 2, different values in Fig. 6). While both variables are defined, the consequences of different $\epsilon$ values are not clearly discussed in the main text and it is not clear how they were chosen.

  3. Clarification on the effect of tension as a growth induced phenomenon: The manuscript explains that non-motile colonies can exhibit tensile stress due to retrograde flow from a proliferative rim toward the center. However, it remains unclear whether this tensile state requires apoptosis in the bulk to create space for inward flow, or whether tension would still arise in the absence of cell death. It would be useful to clarify the different contributions of growth, apoptosis and adhesion to the tensile forces. This is particularly important since one of the main studies on tensile growth cited in the introduction (Ref. [9]) does not consider apoptosis, and the text as written gives the impression that this is one of the results that can now also be explained without motility.

Minor comments

  1. Although the methods mention that the quantities with an asterisk are normalized, this does not become clear in the main text and the different versions with and without asterisk are often mixed (for example, in the caption of Fig. 1, or in Figure 2 caption vs. the plot). The main text should be self-contained in terms of the significance/values of the parameters.

  2. Clarification of coordinate conventions and boundary conditions in the 1D model: In the continuum theory, the quasi-1D colony appears to span from $-x_0$ to $x_0$, with the boundary condition $P(x_0) = 0$ indicating that the colony edge lies at $x = \pm x_0$. However, in the plots (e.g., Fig. 1,3), the pressure appears to vanish at $x = 0$, suggesting a possible mismatch between the theoretical coordinate system and the plotted frame. It would be helpful if the authors could clarify whether the origin in the plots corresponds to the colony center or edge.

  3. The way it is written, negative homeostatic pressure associated with a "negative bulk growth rate at zero pressure" sounds contradictory to the existence of surface growth (because one might intuitively think that the free surface is at zero pressure or even positive due to surface tension). It would be useful to make it more explicit why surface growth still exists.

  4. line 132: Why is this a reasonable assumption for the shape of the motility force density? Or is this purely based on data (specific ref. to SI figure)?

  5. lines 151-153: It is not clear how the different contributions are calculated/measured (numerically/analytically?).

  6. Can the quadratic dependence of tension/expansion speed on $v_0$ be seen more explicitly analytically? Also, in the caption of Fig. 5, it would be good to state the form of the fit, so the conclusion about the quadratic dependence is justified.

  7. Fig. 4b, is there a theory prediction for the velocity as well in analogy to Fig. 3a (or state why it is not shown)?

  8. How are the boundary cells in section 4 defined?

  9. What is $\gamma$ in line 240? Is it the same as $\gamma_D$? $\gamma$ also used in the continuum model.

  10. What is the parameter $B^*$ in the caption of Fig. S10? Is it supposed to be $G^*$?

Typos

  • Inconsistent capitalization for "A(a)ctive B(b)rownian P(p)article".
  • line 45: its $\rightarrow$ it is
  • line 66: optained (also Fig. 3 caption)
  • line 67: capitalized "We", also consider inserting "local" before "growth rate"
  • several instances of "boudnary"
  • line 105: Missing space after "Here,"
  • line 118: capitalized "See" and extra space
  • line 119: "smal"
  • line 167: capitalized "See"
  • line 203: "shortcommings"
  • line 205: "usefull"
  • line 209, line 223: "comparrison"

Recommendation

Ask for minor revision

  • validity: high
  • significance: high
  • originality: high
  • clarity: ok
  • formatting: good
  • grammar: good

Author:  Jens Elgeti  on 2025-07-29  [id 5690]

(in reply to Report 2 on 2025-07-01)
Category:
answer to question

The manuscript by Yao and Elgeti investigates the emergence of mechanical tension in expanding cell colonies growing on substrates, challenging the conventional assumption that tension arises solely from outward-directed cellular motility. Using a combination of particle-based tissue-growth simulations and analytical modeling, the authors demonstrate that tension can spontaneously develop even without motility, driven by mechanical boundary effects promoting growth near colony edges. Incorporating a minimal motility model further amplifies tension and reproduces experimental observations, such as finger-like protrusions at colony fronts. Their findings provide a novel framework for distinguishing different mechanical growth regimes and predict measurable retrograde cellular flows, offering new interpretations for experimental systems. The manuscript is generally well-written and provides interesting perspectives. However, there are some issues regarding presentation and the interpretation of the results which should be addressed before publication.

We kindly thank the referee for carefully reading our manuscript and providing constructive feedback. We respond point by point below.

Disclaimer: We could not find the supplement on SciPost and looked at the arXiv version of the supplement assuming it would be similar.

We had uploaded the SI to Zenodo, and provided a temporary access link in the manuscript. Unfortunately, due to delays on our side, the access had timed out review. We now published our SI on Zenodo, providing permanent DOI based access.

Major comments

  1. Definition of the model and initial presentation of effects: While the model might be very well-known and intuitive for the authors, the beginning of the manuscript is not easy to digest for outside readers: While a few model components are mentioned in the main text, the entire definition is outsourced to methods and the supplement. The main text should at least give the necessary intuition regarding all relevant model mechanisms (apoptosis, adhesion, how does homeostasis arise in this model?) before jumping to the first results (around line 39). In addition, the presentation of the initial results is a bit superficial (Fig. 1), it is not clear how the pressure is measured (although explained in the supplement), etc. It might help to first finish a slightly more expanded model introduction and then switch to a proper presentation of the results.

We agree with the referee that it is difficult to fully grasp the results of figure 1 when first mentioned in the text. Indeed, one basically has to read the manuscript to the end to understand it fully. Our intention is here to give the reader an impression of what to expect. The idea is to give the final result in the introduction, to motivate and help the reader understand, by showcasing what the goal of the work is. We believe it is more difficult to understand a model and follow a manuscript, without a clear understanding of the intention, especially for a broader audience. Naturally, this comes at the price the referee points out. We have reworked the introduction a bit in order to mitigate the disadvantages that the referee has pointed out by explicitly pointing to later parts of the manuscript and expanding the model description.

  1. Model transparency and choice of main parameters: The significance of certain modeling details is not always clear. If we understand correctly, the growth force in Refs. 8 (on which the model is based) and 9 is not constant but depends on the distance between cell particles. Also, a different kind of repulsion is used. What is the rationale behind these changes? Do they alter the results?

The referee is right to point out that we use different forms of forces in this paper than in the previous researches that this paper is based on. Growth force: In the original work, the force decreases with the intracell particle distance. This was due to a misconception that this is required for pressure dependent homeostasis to appear. Now we have realized this is not needed. In the spirit of minimal assumptions, we find a constant force simpler than a distance dependent one. Additionally, the original distance dependent form could cause growth to be stuck at a particular distance in rare cases. After plenty of trials and testing, we chose this simpler form. The general mesoscopic growth properties of the system however are un-altered. Interaction force: The change here is more important, and stems from the goal to combine the 2PG model with the model from Sarkar2021[16]. The key result from Sarkar2021 is that, besides short range repulsion and long range attraction, an intermediate, force-free region is needed to create liquid-vacuum coexistence. We thus switched to the interaction forces from Sarkar2021. This change of form will affect detailed mechanical properties, the exact value of quantities like the homeostatic pressure, but again will not affect the general growth behavior. In addition, LJ interaction is just more familiar to the audience. We modified the text to make the changes more clear and to convey the reasoning behind these changes.

Moreover, the manuscript uses the microscopic parameters growth force G and adhesion strength $\epsilon$ to explore different growth regimes ($\epsilon^*=2$ in Fig. 1, $\epsilon^*=1$ in Fig. 2, different values in Fig. 6). While both variables are defined, the consequences of different $\epsilon$ values are not clearly discussed in the main text and it is not clear how they were chosen.

Indeed, we did not discuss the consequence of changing $\epsilon$. From what we have seen, the most important effects are twofold:

a) $\epsilon$ reduces growth, i.e. for the same growth force, a larger $\epsilon$ will cause the colony to have smaller homeostatic pressure. However, at the same time, it increases the surface growth effect, leading to more tension at the phase boundary between phase I and II.

b) $\epsilon$ caps the maximum tension a tissue can support without fracturing. The smaller $\epsilon$, the larger the “scattered Phase” (Phase 4)

Furthermore, $\epsilon$ also has minor effects on the fluidity and the response coefficient $\kappa$ of the system. So we displayed two $\epsilon$ values in the figures to showcase that colonies with different $\epsilon$ in general behave alike other than the difference in homeostatic pressure and supported tension.

In order to discuss the role of $\epsilon$ more clearly in the manuscript, we added a curve for $\epsilon^*=2$ to Fig.5(b), and explained the points above in the main text.

  1. Clarification on the effect of tension as a growth induced phenomenon: The manuscript explains that non-motile colonies can exhibit tensile stress due to retrograde flow from a proliferative rim toward the center. However, it remains unclear whether this tensile state requires apoptosis in the bulk to create space for inward flow, or whether tension would still arise in the absence of cell death. It would be useful to clarify the different contributions of growth, apoptosis and adhesion to the tensile forces. This is particularly important since one of the main studies on tensile growth cited in the introduction (Ref. [9]) does not consider apoptosis, and the text as written gives the impression that this is one of the results that can now also be explained without motility.

Indeed, apoptosis is necessary for a non-motile colony to be under tension, as the homeostatic pressure, which is the pressure at the center of the colony, cannot be negative without apoptosis. Ref.[9], using the same model in essence, can only generate tension inside the colony by motility as a transient effect of elasticity (motility pulls colony outwards), and will eventually reach a positive homeostatic pressure in the center in the long time limit. Now we have emphasized the requirement of apoptosis in discussion.

The contributions of growth, apoptosis and adhesion to tension are complicated. For example, even though apoptosis is required for tension, larger apoptosis rates does not necessarily correspond to (much) higher tension because it creates more free space, rendering the tissue more liquid, and thus increasing divisions to compensate for the additional death. Inspired by the question of the referee, we added a figure to SI showing how these factors contribute to the homeostatic pressure to show their effects.

Minor comments

  1. Although the methods mention that the quantities with an asterisk are normalized, this does not become clear in the main text and the different versions with and without asterisk are often mixed (for example, in the caption of Fig. 1, or in Figure 2 caption vs. the plot). The main text should be self-contained in terms of the significance/values of the parameters.

We are sorry for the inconsistency of the reduced parameters. We corrected them in the manuscript and explained their use earlier in the manuscript.

  1. Clarification of coordinate conventions and boundary conditions in the 1D model: In the continuum theory, the quasi-1D colony appears to span from −x0 to x0, with the boundary condition P(x0)=0 indicating that the colony edge lies at x=±x0. However, in the plots (e.g., Fig. 1,3), the pressure appears to vanish at x=0, suggesting a possible mismatch between the theoretical coordinate system and the plotted frame. It would be helpful if the authors could clarify whether the origin in the plots corresponds to the colony center or edge.

The referee is right to point out this discrepancy of the surface position. We have clarified this in the caption of Fig.1,3 and 4.

  1. The way it is written, negative homeostatic pressure associated with a "negative bulk growth rate at zero pressure" sounds contradictory to the existence of surface growth (because one might intuitively think that the free surface is at zero pressure or even positive due to surface tension). It would be useful to make it more explicit why surface growth still exists.

Indeed, surface growth is not a simple consequence of pressure. It comes from the free space at the surface and subsequently the ease to push away surroundings to gain space. Indeed, we speculate that this additional growth is due to the reduction in the strain energy of a strain dipole near a free surface. However, as this is speculation and has not been thoroughly analyzed to our knowledge, we do not want to discuss this point in detail at this moment. We explicitly add surface growth as an addition to pressure regulated growth in Eq.1, as also done in previous works [1,2]. Ref. [25] in particular discusses the role of this surface growth for negative homeostatic pressure. We emphasize now clearly this additional, pressure independent, contribution to growth.

  1. line 132: Why is this a reasonable assumption for the shape of the motility force density? Or is this purely based on data (specific ref. to SI figure)?

Indeed, it is a purely phenomenological assumption based on the simulation data. We do not know the dynamics of the polarization profile on the collective level. We make this clearer to the reader now by emphasizing it's phenomenological.

  1. lines 151-153: It is not clear how the different contributions are calculated/measured (numerically/analytically?).

The referee is right to point out that it is not clear how these are measured. We modified this part to emphasize that these contributions are directly measured from simulations.

  1. Can the quadratic dependence of tension/expansion speed on v0 be seen more explicitly analytically?

For now, we have no analytic explanation of the quadratic dependence except the symmetry argument presented in the manuscript: Because a tissue with $v_0$ or $-v_0$ is identical, the lowest order must be quadratic in $v_0$. The nature of this behavior is up to future researches to study. As our raw data is freely available, it will be a valuable source for researchers working on this question.

Also, in the caption of Fig. 5, it would be good to state the form of the fit, so the conclusion about the quadratic dependence is justified.

We agree. We added the quadratic form to the caption of Fig.5.

  1. Fig. 4b, is there a theory prediction for the velocity as well in analogy to Fig. 3a (or state why it is not shown)?

We now plot the theory prediction for the velocity profile in Fig.4(b).

  1. How are the boundary cells in section 4 defined?

In brief, we used k-d tree to determine the number of neighbor cells of each cell and define boundary cells as those with a neighbor cell number below a certain threshold. The method is explained in Supplementary Section S5. This should have been in the main text. We added "See S5 for the determination of boundary cells" in the caption of Fig.6.

  1. What is $\gamma$ in line 240? Is it the same as $\gamma_D$? $\gamma$ also used in the continuum model.

We are sorry for the typos. Indeed it should be $\gamma_D$. We corrected it.

  1. What is the parameter $B^*$ in the caption of Fig. S10? Is it supposed to be $G^*$?

We are sorry for the outdated version of the SI on the arxiv. It has been corrected now.

Typos

We are sorry for the typos in the manuscript and thank the referee for pointing them out. They have been corrected.

---

## Round 2 · Referee Report · Saraswat Bhattacharyya (Referee 1) · 2025-8-28

Report

I thank the authors for addressing my concerns and comments. I am happy to recommend this paper for publication.

Recommendation

Publish (easily meets expectations and criteria for this Journal; among top 50%)

---

## Round 2 · Referee Report · Anonymous (Referee 2) · 2025-9-11

Report

We thank the authors for their thorough responses to our comments and the improvements made to the manuscript, which we now recommend for publication.

Recommendation

Publish (easily meets expectations and criteria for this Journal; among top 50%)

---

## Round 2 · List of Changes

(see referee responses)

---

## Editorial Decision

published